# Unpacking climate effects on boreal tree growth: An analysis of treering widths across temperature and soil moisture gradients

Andreas Lundgren<sup>1</sup>, Joachim Strengbom<sup>1</sup>, Johannes Edvardsson<sup>2</sup>, Gustaf Granath<sup>3</sup>

<sup>1</sup>Department of Ecology, Swedish University of Agricultural Sciences, 75007, Sweden

<sup>2</sup>Laboratory of Wood Anatomy and Dendrochronology, Department of Geology, Lund University, 22362, Sweden

<sup>3</sup>Department of Ecology and Genetics, Uppsala University, 75236, Sweden

Correspondence to: Andreas Lundgren (andreas.lundgren@slu.se)

**Abstract.** The effect of climate change on tree growth in boreal forests is likely mediated by local climate conditions and species-specific responses that vary according to differences in traits. Here, we assess species-specific tree growth responses to climate along gradients of mean annual temperature and soil moisture.

We assessed growth-climate relationships by using tree-ring width data in Norway spruce (*Picea abies*) and Scots pine (*Pinus sylvestris*) from the Swedish National Forest Inventory in relation to climatological data along gradients in mean annual temperature and soil moisture.

Trees growing in warmer areas responded more negatively to high temperature and more positively to high precipitation. Site-specific soil moisture only showed an effect on the growth responses in areas of high mean annual temperature. The growth-climate response differed between the species; specifically, the growth response to high temperature varied more along the gradient of mean annual temperature for *P. abies* than for *P. sylvestris*. Growth responses to extreme weather events did not deviate from non-extreme events along the climatic gradients.

Our study suggests that tree growth responses to climate change will depend on tree species and site-specific climate conditions. In warmer areas, high soil moisture may mitigate the adverse effects of warming on tree growth mainly for *P. abies*. In colder areas, *P. abies* is likely to benefit more from warming than *P. sylvestris*. Although the matching between extreme tree growth and extreme temperature or precipitation years was consistently higher than expected if the two variables were independent, an extreme year is unlikely to cause a tree growth response that markedly diverges from predictions based on linear relationships. Thus, the amplification of negative growth-climate responses during extreme years is likely of limited importance for long-term growth, as such events are inherently rare. Nevertheless, extreme years may influence forest productivity by affecting tree mortality, an aspect that was beyond the scope of this study. In the face of climate change, our results emphasize that forest management should consider site-specific climate conditions and species differences to sustain future forest productivity.

## 1 Introduction

35

40

The boreal forests constitute 27% of the global forest area (FAO, 2020) and play a significant role for global carbon storage (Pan et al., 2024). Presently, about two-thirds of the boreal forest area is managed (Gauthier et al., 2015), and since the growing forest stock in the boreal region as of 2020 corresponds to 24% of the global stock (FAO, 2024), its tree growth also represents a significant economic value. However, under global warming, the delivery of these ecosystem services is at risk. Boreal forests and their tree growth may be particularly vulnerable (Babst et al., 2019), as the rate and impact of increasing temperature are predicted to be more pronounced at higher latitudes (IPCC et al., 2019). This has raised concerns that the boreal biome's capacity to assimilate and sequester carbon could be at stake by the mid of this century (Rao et al., 2023). Moreover, there are already indications that global warming has reduced the biome's geographical range, with forest contraction in its southern areas and limited expansion in the north (Rotbarth et al., 2023). Given its global importance, predicting the response of boreal forests to global warming becomes of vital importance.

With ongoing and projected warming of the climate (IPCC, 2023), tree growth is expected to increase in the boreal region (Kauppi et al., 2014; Pau et al., 2022), attributable to the positive relationship between temperature and photosynthetic activity (Saxe et al., 2001). However, increasing temperature also cause an increase in atmospheric vapour pressure deficit (VPD) which has the potential to reduce plant growth rates (Novick et al., 2024; López et al., 2021; Yuan et al., 2019). The temperature-growth response is complex as growth may increase with rising temperatures to a certain threshold, beyond which high VPD leads to a reduction in growth (Grossiord et al., 2020). Assuming that tree growth has a temperature optimum (Sendall et al., 2015), growth in colder regions is expected to increase under global warming, whereas growth in warmer regions may decline if the temperature exceeds the optimum. This effect can be further amplified by a longer growing season, as anticipated with an extension of the snow free period (Gustafson et al., 2024), as rising temperatures are likely to enhance tree growth in colder regions but limit growth in warmer areas due to increased water stress (Gao et al., 2022). Therefore, it can be hypothesized that a warmer climate is likely to increase tree growth in relatively cold regions, but decrease tree growth in warmer regions.

Tree growth is often influenced by small temperature changes (Reich et al., 2022), and growth declines may act as early warning signals for more severe impacts, such as forest dieback (Popa et al., 2024; Gazol et al 2020). In addition, warming may also increase the frequency and intensity of extreme disturbances (Gustafson et al., 2024; Gauthier et al., 2015). Extreme events such as droughts may have a far more detrimental effect than the direct effect of increased temperature (Peng et al., 2011), and sometimes affect the tree growth over multiple years (Babst et al., 2012). Years characterized by unusually high temperature may also increase tree mortality (Peng et al., 2011) due to hydraulic failure or prolonged photosynthetic inactivity following stomatal closure in response to the drought conditions (Klein et al., 2014) and therefore decrease forest productivity (Barber et al., 2000). Furthermore, species that normally show a positive growth response to elevated temperatures may exhibit the opposite response and become negatively affected when temperatures become excessively high (Reich et al., 2022). Such extreme weather events are predicted with high certainty to become more frequent in the boreal region (IPCC, 2021). For

example, an increased global mean temperature of 1.5 °C is predicted to increase the frequency of extreme temperature events by 4.1 per decade compared with levels prior to 1900 (IPCC, 2021). The high probability of more frequent extreme events makes it vital to understand if droughts and heat spells exacerbate the overall effects of climate change on tree growth, and if the impact of such extreme events can be predicted based on the same relationship as changes in mean temperature.

65

Local factors strongly influence tree growth responses to rising temperatures, causing substantial variation across geographical locations (D'Orangeville et al., 2016; Ols et al., 2018; Pedlar & McKenney, 2017; Perret et al., 2024). Accounting for such local effects is essential for accurately predicting the impacts of climate change on tree growth. For instance, the potential adverse impacts of rising temperature on tree growth may be mitigated by soil moisture conditions, as studies have shown that the positive correlations between tree growth and temperature is enhanced at sites with high soil moisture (Pau et al., 2022). Similarly, increased temperature may aggravate the effects of drought on tree growth more in dry sites (Gagne et al., 2020). The opposing influences of soil moisture might have restrained the influence of rising temperature on tree growth in the boreal region. Nonetheless, recent findings of reduced water content in boreal forest vegetation (Wang et al., 2023) could indicate a diminishing mitigating effect of soil moisture. Correspondingly, it has been suggested that alterations in soil moisture alone may have neutralized the potential advantages of warming on forest growth in certain areas (D'Orangeville et al., 2016). Therefore, to reliably predict future tree growth, it is important to understand how site-specific characteristics, such as soil moisture, interact with increasing temperature.

Tree species may show different responses to warming, and the response to increasing temperature is expected to vary even among different species of conifers (Jevšenak & Saražin, 2023; Klein, 2014). While both *Picea abies* and *Pinus sylvestris*, the two most common tree species in the Fennoscandian boreal forest (SLU, 2025b), adopt a relatively isohydric strategy (Leo et al., 2014), *P. abies* has often been found to be more sensitive to drought than *P. sylvestris* (Gutierrez Lopez et al., 2021; Treml et al., 2022). This difference may be partly explained by the higher root-to-leaf ratios in *P. sylvestris* which likely provide greater access to water during drought (Helmisaari et al., 2007). However, species-specific drought responses can also be shaped by site-specific environmental conditions (Feng et al., 2019). Projections of 21st-century tree growth suggest that *P. sylvestris* will be more strongly influenced by local climate, with growth increasing in colder regions but declining in warmer ones, whereas *P. abies* is expected to show a more uniform growth response (Martinez del Castillo et al., 2024). Furthermore, since *P. sylvestris* generally thrives on drier soils than *P. abies* (Sutinen and Middleton, 2020), it may be less affected by adverse conditions in dry areas. Variation in sensitivity and responses to local conditions among tree species may lead to altered species composition, potentially impacting the ecosystem services they provide, such as wood production and carbon sequestration (Huuskonen et al., 2021). Hence, increased understanding of how tree growth responses interact with local conditions is crucial not only for comprehending ecological responses to global warming but also for practical implications, e.g. for management practices.

Numerous studies have shown an impact of ongoing climate change on forest growth (Aldea et al., 2024; Babst et al., 2019; Boisvenue & Running, 2006; Perret et al., 2024; Popa et al., 2024). However, significant uncertainties remain regarding the interaction between local factors and growth-climate relationships. Without knowledge of how growth-climate responses

differ among species and site-specific climate, we risk extrapolating unrepresentative growth trends based on one area or species to those that are governed by different factors. To address this knowledge gap, we quantified the impact of temperature and precipitation on the growth of two boreal tree species, namely Norwegian spruce (*Picea abies*) and Scots pine (*Pinus sylvestris*), along a 1450 km climate gradient in Sweden. Specifically, we tested five hypotheses. (H1) Tree growth response to high temperature is negative in regions with relatively high mean annual temperature, i.e. the southern range of the studied gradient, but becomes increasingly positive in colder regions. Furthermore, we tested how the tree growth response to other climate predictors (precipitation, VPD, and SPEI) varied along the mean annual temperature gradient. (H2) Increasing soil moisture, as a result of local topography, mitigates the negative impacts of increased temperature and lower precipitation on tree growth. (H3) The growth response and its interaction with mean annual temperature and soil moisture will differ among tree species. Specifically, we anticipate that *P. sylvestris* has a more positive response to increased temperature in relatively cold areas and more negative response to increased temperature in dry areas compared to *P. abies*. Furthermore, we anticipate that *P. sylvestris* has a more positive response to increased temperature in dry areas compared to *P. abies*. (H4) Extreme years in terms of high temperature, or low precipitation, coincide with years of exceptionally low tree growth more often in regions of high mean annual temperature and precipitation extremes.

## 2 Methods

## 2.1 Site description

For our study, we used data on tree radial growth and climatic variables from the entire range of the boreal forest region in Sweden. The forests are dominated by P. abies and P. sylvestries which constitute ~40% each of the total growing stock (SLU, 2025a). Within the Swedish boreal forest there are some regional differences, with P. sylvestris being more common in the north than in the south, constituting ~50% and ~30% of the total growing stock in the northernmost and the southernmost regions, respectively. Conversely, P. abies is more common in the south than in the north, constituting ~30% and ~40% of the total growing stock in the northernmost and the southernmost regions, respectively.

The mean annual temperature in our study area during 1991-2020 ranged from 10 °C in the south to -2 °C in the north (SMHI, 2025). The mean annual temperature has increased by 0.5-2 °C compared to the period of 1961-1990. The temperature increase has not shown a clear north-south gradient. The mean annual precipitation during 1991-2020 ranged from 400 to 1200 mm, although without a clear difference along the north-south gradient. The mean annual precipitation has increased by 20-200 mm compared to the period of 1961-1990, with the greatest increase in the south-west. A few relatively small areas in the middle of our latitudinal range have experienced a reduced mean annual precipitation of 20-60 mm. The annual maximum snow cover during 1991-2020 ranged from <10 cm in the south to 200 cm in the north. The maximum snow cover has decreased by 5-25 cm compared to the period of 1961-1990 in the south and in the middle of the study area, while remaining stable or experiencing a slight increase in the northernmost parts.

# 2.2 Tree core data

In order to explore tree growth responses to climatic variables, we used radial growth data extracted from tree cores collected by the Swedish National Forest Inventory (NFI) between 2018 and 2022 at sites throughout Sweden (SLU, 2022). Similar radial growth data collected by the Swedish NFI has previously been used to examine the impact of oceanic dynamics (Ols et al., 2018) and drought (Aldea et al., 2023) on tree growth. The NFI study areas consist of 0.25 - 1 km<sup>2</sup> squares that are systematically chosen through a grid design (Ranneby et al., 1987). Within each area, 6-10 survey plots are randomly distributed. In each plot, all trees within a 7 m radius are identified and the diameter at breast height (DBH) is measured. One to three trees are chosen for a single coring through unequal probability systematic sampling based on probabilities proportional to basal area (see detailed explanation in Fridman et al., 2014). The tree cores are cut with a microtome, treated with zinc paste and tree ring widths (TRW) are analyzed through a camera microscope. TRW are measured for the latest sixty annual growth rings of each tree core. Where < 60 annual rings are detected. TRWs for all rings are measured. To minimize significant agerelated variations, we exclusively analyzed trees aged > 40 years in our study. Furthermore, to limit our study to trees growing in productive forests (mean annual tree growth rate > 1 m<sup>3</sup> ha<sup>-1</sup>), we excluded trees growing in wetlands, and on peat soils (organic layer > 30 cm) from the analysis. Prior to analyses, we created ring-width indices (RWI) by detrending each tree's TRW series with a spline function through the R package dplR (Bunn et al., 2023), using a 50% cut-off after 30 years. Nine time-series produced unreasonable RWI with the last year's RWI reaching a value magnitudes higher than the rest and were removed from further analysis. We conducted a quality assessment of the RWI series by performing inter-series correlations using the R package dplR (Bunn et al., 2023) for all trees of the same species within the same study area. We excluded trees with an inter-series correlation below the critical confidence level of 0.3281 from further analyses. The choice of confidence level was based on what is commonly used in the dendrochronological software COFECHA (Holmes, 1983). Further, we excluded all study areas containing 

Figure 1. Map of the National Forest Inventory (NFI) sample sites used in this study. (A) shows geographical distribution of plots along a mean annual temperature (MAT) gradient. (B) shows latitudinal distribution of plots along a soil moisture (SMI) gradient, where the x-axis indicates the number of plots per 0.1 latitude and the color gradient indicates the probability of a site being classified as "wet" in increments of 20% (yellow = 0-20%, light green = 21-40%, dark green = 41-60%, blue = 61-80%, purple = 81-100%). For more information on the SMI classification, see section 2.3.

## 2.3 Climate and soil moisture data

Data on mean, maximum, and minimum daily temperature, mean daily relative humidity and daily precipitation sum for the period 1961-2018 were retrieved for each study site from nationwide modeled data on a 2.5 km resolution grid (Andersson et al., 2021). In addition, we calculated a Standardized Precipitation-Evapotranspiration Index (SPEI) for each month using the R package SPEI (Beguería & Vicente-Serrano, 2023). A necessary component in the SPEI calculation is potential evapotranspiration, which was calculated using the Hargreaves function (Vicente-Serrano et al., 2010), where net radiation was inferred from latitude. SPEI centers around 0, where negative values indicate drier conditions and positive values indicate wetter conditions. Furthermore, we calculated the vapour pressure deficit (VPD) through relative humidity and saturated

vapour pressure based on mean temperature (Howell & Dusek, 1995). We also calculated the number of days that the threshold of VPD > 1.5 kPa had been exceeded by calculating the VPD based on maximum daily temperature. The threshold level was justified as it has been proposed as the value at which stomata closure is triggered (Kurjak et al., 2012). However, the number of days exceeding the threshold correlated strongly with the mean growing season VPD, and we therefore removed this variable from further analysis. Prior to analysis, data on mean temperature, precipitation sum, SPEI, and VPD were aggregated to growing season means, or sums for precipitation. The growing season for each site and year was calculated by assuming that the growing season starts at the first of four consecutive days with a daily mean temperature > 5 °C and ends after four consecutive days with a daily mean temperature > 5 °C and ends after four consecutive days with a daily mean temperature > 5 °C (SMHI, 2011). To avoid anomalies in growing season length, a limit was set where the growing season could not start before March 1st and could not end until after August 31st. Prior to analyses, we followed the recommendation of Ols et al (Ols et al., 2023) and detrended each climatic time-series data similarly to the detrending we used for TRW. As some SPEI values were negative, and the detrending function we used cannot handle this, we added the lowest value from all SPEI values so that they were all be positive prior to detrending.

Data on soil moisture for each individual site were retrieved from a Swedish soil moisture model that is largely based on topography and ground water data, and validated by NFI permanent plots (Ågren et al., 2021). The modeled data consist of values ranging from 0 to 100 at a 2 m resolution raster grid, where values indicate the probability of being classified as the "wet" category in the NFI inventory field plots. The NFI classification of "wet" indicates water-saturated soil with visible surface water originating from groundwater (SLU, 2025b). Using the modeled soil moisture data, we calculated a soil moisture index (SMI) based on mean soil moisture values in a buffer of 25 m radius around individual trees using zonal statistics in QGIS (QGIS Association, 2021).

## 2.4 Data analysis

To test our first three hypotheses on how tree growth responses to climatic variables vary along a MAT gradient, an SMI gradient, and between *P. abies* and *P. sylvestris*, we fitted linear mixed models using the R package nlme (Pinheiro et al., 2025). We fitted one model for each climatic variable (temperature, precipitation, SPEI, and VPD) to test the four-way interaction effect of the climatic variable x MAT x SMI x tree species on RWI, as well as the effects of all lower order terms. To account for spatial autocorrelation within the models, we added plot as a random variable. We also estimated variograms at multiple time points to explore the possibilities to include a more complex spatial process. However, we did not find any shape in these variograms, regardless of the model used (exponential, spherical, Gaussian) and this approach was abandoned. To account for temporal autocorrelations, we added a continuous-time autoregressive factor of order 1 (AR1). With this approach, the interaction term between each respective climatic variable and MAT, SMI, or species, represents how the tree growth response to the climatic variable varies along the gradients and between species.

To test our fourth hypothesis that extremely warm or dry years coincide with years of low tree growth more often in regions of high MAT or low SMI, we calculated coincidence rates and tested these along the MAT and SMI gradients as well as between species. To calculate the coincidence rates, we counted the number of extreme events that the RWI series and

temperature series have in common (Rammig et al., 2015). We identified extreme years in terms of high temperature as those inside the top 10% of mean growing season temperature for each site. We then identified the years with the lowest 10% of RWI for each tree. We counted the number of coincidences for each tree as the number of years that appeared both among the 10% highest temperature years and 10% lowest RWI years. Finally, we normalized the coincidence rate for each tree by dividing the number of coincidences with the total number of extreme years (i.e. 10% of the tree's time series). With this approach, any coincidence rate above 0.1 indicates that years of extremely high temperature coincide with years of extremely low growth on more occasions than can be expected if the two variables are uncorrelated. Any coincidence rate below 0.1 indicates that these years coincide more rarely than can be expected from independence, i.e., a negative correlation between the two variables. We then used linear mixed models to test the three-way interaction effect of MAT x SMI x tree species on the coincidence rates, as well as the effects of all lower order terms. To account for spatial autocorrelation within the models, we added plot as a random variable. We used the same approach for testing coincidence rates with extreme years in terms of low precipitation.

To test our fifth hypothesis that tree growth in regions of high MAT or low SMI is less resistant to extremes, we calculated resistance values for the extreme years and tested these along the MAT and SMI gradients as well as between species. Using the same years of high temperature as those used for calculating the coincidence rates, we determined the resistance values (Lloret et al., 2011) for each tree by dividing the RWI of the target year by the mean RWI of the three preceding years. With this approach, any value < 1 indicates a decrease in growth during the extreme years, and any value > 1 indicates an increase in growth during the extreme years. We then used linear mixed models to test the three-way interaction effect of MAT x SMI x tree species on the resistance values, as well as the effects of all lower order terms. Since we used 10% of each tree's time series (i.e. several years per tree), we added tree identity nested within plot as a random variable to make up for within-tree dependency.

For all linear mixed effects models, we tested the fixed effects of the models with Type II Wald Chi-square tests from the R package car (Fox & Weisberg, 2019).

All data analyses were conducted in R (R Core Team, 2021).

## 3. Results

## 3.1 Influence of mean annual temperature, soil moisture, and tree species on RWI

Averaged across the entire dataset, RWI increased with increasing temperature and VPD but decreased with increasing precipitation and SPEI (Table A1-A2). However, the direction of these effects was highly dependent on site-specific characteristics. In support of our first hypothesis (H1), we found that the tree growth response to temperature was negative in regions with relatively high MAT, but became increasingly positive in colder regions (Fig. 2; Table 1). The growth response to VPD followed a similar pattern throughout the MAT gradient. The growth response to precipitation and SPEI became increasingly positive with increasing MAT (Fig. A2; Table 1).

Figure 2. Model output of the effects of temperature on RWI (ring width index) when separated between low (10% quantile; blue lines), median (black lines), and high (90% quantile; red lines) MAT (mean annual temperature). Lines show the differences between low (10% quantile; solid line), and high (90% quantile; dashed line) SMI (soil moisture index). Facets show differences between species (left = P. abies; right = P. sylvestris). Note that quantiles represent different values for the different species. SMI values for P. abies (and P. sylvestris in parentheses) are 2.7 (1.9) and 85 (87), for low and high values, respectively. MAT values are 0.9 (1.1), 3.4 (3.7), and 7.4 (7.4), for low, median, and high values, respectively.

Table 1. Output ( $\chi^2$ - and p-values) of linear mixed effects models testing the interaction effects of each respective climatic variable (temperature, VPD, precipitation, and SPEI during the growing season), mean annual temperature (MAT), soil moisture (SMI), and tree species (TS), on RWI. Sample size of each model is 1979. Note that main effects are not shown (see Table A1-A4 for main and random effects values of each model).

| Interaction term             | $\chi^2$ | p      |
|------------------------------|----------|--------|
| Temperature x MAT            | 2669     | 

Figure 3. Model output of the effects of MAT on rate of coincidence between years of low growth and high temperature (purple line) or low precipitation (green line). Facets show the differences between low (10% quantile; left graphs), median (middle graphs), and high (90% quantile; right graphs) SMI, as well as between species (upper = P. abies; lower = P. sylvestris). Note that quantiles represent different values for the different species. SMI values for P. abies (and P. sylvestris in parentheses) are 2.7 (1.9), 27 (26), and 85 (87), for low, median, and high values respectively, Dashed line represents the 0.1 threshold, above which the years coincide more often than can be expected from random coincidences.

Table 2. Output ( $\chi^2$ - and p-values) of linear mixed effects models testing the effects of mean annual temperature (MAT), soil moisture (SMI), and tree species (TS), on coincidence rates between the years of 10% lowest RWI and years of either 10% highest temperature or 10% lowest precipitation. The sample size of both models is 1979.

| Term           | High temperature |        | Low precipitation |        |
|----------------|------------------|--------|-------------------|--------|
|                | $\chi^2$         | p      | $\chi^2$          | p      |
| SMI            | 3.28             | 0.07   | 3.14              | 0.08   |
| TS             | 244              | 

Figure 4. Model output of the effects of MAT on tree growth resistance to years of high temperature (purple line) or low precipitation (green line). Facets show the differences between low (10% quantile; left graphs), median (middle graphs), and high (90% quantile; right graphs) SMI, as well as between species (upper = P. abies; lower = P. sylvestris). Note that quantiles represent different values for the different species. SMI values for P. abies (and P. sylvestris in parentheses) are 2.7 (1.9), 27 (26), and 85 (87), for low, median, and high values respectively, Dashed line represents the 1.0 threshold, below which the extremes have a negative impact on tree growth.

Table 3. Output ( $\chi^2$ - and p-values) of linear mixed effects models testing the effects of mean annual temperature (MAT), soil moisture (SMI), and tree species (TS), on resistance values to the years of either 10% highest temperature or 10% lowest precipitation. The sample size of both models is 1979.

|                | High temperature |        | Low precipitation |        |
|----------------|------------------|--------|-------------------|--------|
|                | $\chi^2$         | p      | $\chi^2$          | p      |
| SMI            | 2.60             | 0.11   | 9.24              | < 0.01 |
| TS             | 344              | < 0.01 | 17.8              | < 0.01 |
| MAT            | 443              | < 0.01 | 320               | < 0.01 |
| SMI x TS       | 0.42             | 0.51   | 0.81              | 0.37   |
| SMI x MAT      | 7.28             | < 0.01 | 6.15              | 0.01   |
| TS x MAT       | 123              | < 0.01 | 126               | < 0.01 |
| SMI x TS x MAT | 7.68             | < 0.01 | 1.28              | 0.26   |

## 4. Discussion

In this study, we examined the regional differences in tree growth responses to climatic variables and found that the growth response to temperature and precipitation is dependent on the site-specific mean annual temperature (MAT). Furthermore, we found that while soil moisture (SMI) has an insignificant intrinsic effect, it can somewhat mitigate negative effects of changing temperature and precipitation on growth in areas of high MAT.

## 4.1 The influence of local temperature on tree growth responses to environmental changes

We found support for our first hypothesis (H1) that the response of tree growth to temperature and precipitation shifts along a gradient of site-specific MAT (Fig. 2; Table 1). In terms of growth responses to temperature, similar results have been observed in earlier studies throughout Europe, North America and northern Asia, showing progressively negative correlations to temperature along gradients of increasing mean growing season temperature (Klesse et al., 2018; Ols et al., 2018), or positive along latitudinal gradients (D'Orangeville et al., 2016; Li et al., 2020). Although boreal forests experience relatively low temperature, similar patterns have also been observed in the tropical (Zuidema et al., 2022) and temperate (Charru et al., 2017) biomes. As photosynthetic rates increase with temperature (Kellomäki & Wang, 1996), this likely underlines the positive growth response to temperature observed in the colder regions of our study. However, in the warmer regions, the positive effects of increased temperature might be outweighed by heat stress resulting from temperatures exceeding an optimum threshold (Gantois, 2022). Such heat-stress will eventually reach a critical point, resulting in tree mortality (Huang et al., 2015). Even if this critical threshold is not surpassed, the interplay between heat stress and other disturbances, such as pest infestations, has the potential to intensify stress levels and induce significant changes well before reaching the critical temperature threshold (Rever et al., 2015). The divergent responses to increased temperature observed in our study can, therefore, have significant implications for the functioning and management decisions of boreal forest ecosystems. Hence, our findings should be considered when developing future management strategies aimed at ensuring the health of these ecosystems. This will be particularly important in parts of the boreal biome where forestry is important for the economy and the implications for tree growth have societal significance.

For precipitation, we found that colder regions respond more negatively to increased precipitation. A potential mechanism of the negative effects of precipitation is that higher amounts of snowfall delays the start of the growing season (D'Orangeville et al., 2016). However, the negative response in our study is based on growing season precipitation sums only, thereby likely excluding snowfall unless a correlation exists between winter and summer precipitation. Another possible explanation for the negative effect of high precipitation on tree growth in the colder regions is that the forests in the north are already near water saturation and excess precipitation causes waterlogging (Laudon et al 2024). The effect of precipitation along our temperature gradient is inconsistent with studies suggesting that precipitation is consistently positively correlated to tree growth, regardless of site-specific temperature or latitude (Li et al., 2020; Restaino et al., 2016; Walker et al., 2015). However, our results are in accordance with studies by D'Orangeville et al. (2016) and Babst et al. (2013) that have

demonstrated an increasingly negative correlation with latitude. The studies conducted by Walker et al. (2015) and D'Orangeville et al. (2016) both investigated black spruce in North America, but reported different results regarding the impact of precipitation. However, the study by Walker et al. (2015) is a comparison of north- and south-facing slopes rather than comparisons over large geographical areas. This suggests that variation in growth responses to precipitation becomes evident at larger spatial scales rather than at smaller scale variations generated by, for example, slope aspect. However, a positive growth response to precipitation across a large spatial scale is not uniform among all tree species, and has not been observed for species in genera such as Larix, Pinus (Li et al., 2020) and Pseudotsuga (Restaino et al., 2016). Thus, the growth responses to precipitation across large spatial scales seem to be complex, and future studies need to confirm whether our results are general or only representative for the Fennoscandian boreal forest. Climate models generally predict increased precipitation across our study area (IPCC, 2023), which might mitigate the negative effect of increasing temperature on tree growth in the warmer parts of the boreal forest. However, the increased precipitation is expected to come as heavy rainfalls rather than a temporally even increase (IPCC, 2021), and the mitigating effect from changed precipitation may therefore be smaller than expected. In fact, a positive tree growth-precipitation correlation can be weakened if the precipitation occurs as infrequent heavy rains (Land et al., 2017). Therefore, while increased precipitation as an average during the growing season is positive in the warmer regions of our study, an increase in precipitation through intermittent heavy rainfall may show less positive effects.

## 4.2 The effect of soil moisture on tree growth responses to environmental changes

Our second hypothesis (H2) was only partly supported, trees' growth response to temperature and precipitation was weakly affected by local soil moisture, and only in warmer regions (Fig. 2; Table 1). This weak effect is surprising as an earlier study from Canada found that trees growing in already wet areas showed a weak response to drought, whereas tree growth in drier areas was reduced to zero during drought (Huang et al., 2015). Notably, we excluded trees growing in wetlands in our study. The inclusions of such trees may have revealed a greater effect of soil moisture on the trees' growth response to drought conditions. However, our results are in line with Lange et al's (2018) finding that the effect of small-scale site-specific conditions is weak in comparison to larger scale climate regimes. Furthermore, Zweifel et al (2006) found that even small amounts of rainfall could offset a negative relationship between soil water deficits and tree growth, and argued that it was the wetting of the crown rather than the soil that provided this benefit. Such an effect might diminish the importance of soil moisture content and explain the surprisingly weak effect that we observed. While the intrinsic effect of SMI in our study was small, it did interact with MAT, such that the negative effects of high temperature was mitigated by high soil moisture values in warmer areas (Fig. 2). This mitigating effect may be the consequence of retained stomatal conductance despite increasing temperature due to higher levels of soil water availability (Novick et al., 2024). Indeed, the photosynthesis of several tree species has previously been found to be more affected by low soil moisture when exposed to warming (Reich et al., 2018). Furthermore, the temperature optimum, beyond which VPD tends to negatively affect trees' growth rates, has been found to occur at higher temperature in areas of wetter soils (Novick et al., 2024). Possibly, only trees growing in warm areas experienced an atmospheric water demand high enough for soil moisture to actually limit the tree growth in our study system. Future climate change may push previously cold areas into warmer states, making the buffering effect of soil moisture relevant in more locations and increasingly critical in the region covered by our study. This may complicate climate change adaptations of forest management, as site-specific soil moisture should be considered in warm areas, while other factors such as MAT or VPD should take precedence when adapting management to climate change in relatively cold areas.

## 4.3 Differences between species

We found contrasting results in regard to our third hypothesis (H3) that the growth response of P. sylvestris would be more 365 dependent on local climatic factors than P. abies (Fig. 2; Table 1). While P. abies showed a generally more positive response to increasing temperature than P. sylvestris, the opposite was true for increasing precipitation where P. sylvestris showed a more positive response compared to P. abies. This is surprising as P. svlvestris has been shown to physiologically benefit more from warming than P. abies (Kivimäenpää et al., 2017). Furthermore, P. sylvestris has a higher root:leaf ratio than P. abies 370 (Helmisaari et al., 2007), which ought to make them less dependent on sufficient precipitation and less detrimentally affected by increasing temperature. However, P. sylvestris may act more isohydric than P. abies, at least based on sapflow responses to drought conditions (Leo et al., 2014). This may explain the differences seen in our study as high temperature would force a stronger growth decline through increased atmospheric water demand, and increasing precipitation may alleviate such stress more for P. sylvestris than for P. abies. Interestingly, we did not find any interactive effects between species and soil moisture, 375 even though they are known to separate their distribution based on soil water contents (Sutinen & Middleton, 2020). On the contrary, the differences in growth-climate responses between our studied species were more dependent on MAT. Hence, our results indicate that climate adaptation in forest management needs to consider the tree species, where the growth of P. abies likely tolerate greater temperature increases, especially in colder regions, whereas P. sylvestris may benefit from increased precipitation in warmer regions.

## 380 4.4 The effect of extreme years on tree growth

Extreme events have been noted to be important drivers of tree growth, where for example, drought events inferred by SPEI or climatic water deficit (Wu et al., 2022), or persistent extreme heat waves (Yang et al., 2023) might cause growth reductions. Extreme weather conditions may also be more influential than changes of the growing season averages for trees in a specific site (Sanginés de Cárcer et al., 2018). We did find support for our fourth and fifth hypotheses (H4 and H5) that climatically extreme years would coincide with weakened tree growth more often in warm areas (Fig. 3; Table 2), and that the resistance to these extremes would follow our MAT and SMI gradients (Fig. 4; Table 3). However, the coincidence rates and resistance values calculated here followed much the same pattern as for the growth response models conducted on the whole dataset. Furthermore, while the mean coincidence rates observed in our study are consistently larger than what we could expect if extreme events and growth were uncorrelated, the rates are relatively low (Zhang et al., 2023). Given these values, it is unlikely that a year of extreme temperature or precipitation sum would have an extreme effect on tree growth that deviates from

predictions based on linear relationships across climate variable space. Therefore, amplification of negative growth-climate responses during extreme years may be of limited importance for long-term growth, as these events are inherently rare. However, the extremes of our studied time series might not be representative of those that may come with further climate change (IPCC, 2021). Furthermore, it is possible that the extremes in our study, based on high or low temperature or precipitation sums during the growing season, are too blunt to capture biologically important extreme weather conditions. It is also important to note that we have studied discrete extreme years, whereas extended periods of extreme conditions rather than single-year extremes may produce the most severe effects for tree growth (Gustafson et al., 2016). Climate change may impact forests in additional ways beyond what we have studied, such as more frequent forest fires and insect outbreaks, which may have greater influence on growth than the predictors we investigated. As our analyses consider only living trees, it is also possible that there are effects of extreme years on mortality rates that we have not captured here. These caveats suggest that our findings that extremes affect growth-climate responses similarly to growing season averages, should be viewed with some caution.

## 5. Conclusion

These results indicate that climate change adaptations in forest management must differ across the boreal biome and consider species selection. We found that warmer areas (higher MAT) in the studied region exhibited more negative growth responses to high temperature and more positive responses to high precipitation. On average, *P. abies* responded more positively to higher temperature than *P. sylvestris*, but the difference between the two species was strongly influenced by local MAT. In the warmer, southern range, sites with high soil moisture can mitigate the negative effects of higher temperatures on *P. abies*. By contrast, in the cooler, northern range, such local variation plays a more limited role; here, increasing temperatures are likely to enhance the growth of *P. abies* but not *P. sylvestris*. Moreover, growth responses to extreme weather events, although sometimes co-occurring with years of low growth, followed similar patterns to those observed during non-extreme conditions along the MAT and SMI gradients. This consistency suggests that climate change-adapted management to mitigate extreme events does not require different considerations than managing for general climate change. Overall, our study suggests that management practices tailored to site-specific and species-specific requirements are crucial to maintaining high tree growth and the overall health of boreal forests.

## Appendix A

## **Effect tables**

Table A1. Output ( $\chi^2$ - and p-values) of linear mixed effects models testing the effects of temperature, mean annual temperature (MAT), soil moisture (SMI), and tree species (TS), on RWI. For main-effect terms, the variance inflation factor (VIF) is given (computed from the r-package *car* on a model containing only the main effects). Model sample size is 4578 RWI nested in 1979 plots. Phi-value of the AR1 temporal autocorrelation = 0.40. Plot intercept variance = 2.77x10<sup>-11</sup>. Tree nested within plot variance = 1.14x10<sup>-11</sup>.

| Term                         | χ <sup>2</sup> | p      | VIF   |
|------------------------------|----------------|--------|-------|
| Temperature                  | 323            | < 0.01 | 1.000 |
| SMI                          | 0.09           | 0.76   | 1.008 |
| TS                           | 0.01           | 0.92   | 1.006 |
| MAT                          | 1.63           | 0.20   | 1.013 |
| Temperature x SMI            | 11.0           | < 0.01 | -     |
| Temperature x TS             | 2558           | < 0.01 | -     |
| SMI x TS                     | < 0.01         | 0.96   | -     |
| Temperature x MAT            | 2669           | < 0.01 | -     |
| SMI x MAT                    | 0.30           | 0.58   | -     |
| TS x MAT                     | 0.29           | 0.59   | -     |
| Temperature x SMI x TS       | 0.03           | 0.87   | -     |
| Temperature x SMI x MAT      | 22.0           | < 0.01 | -     |
| Temperature x TS x MAT       | 618            | < 0.01 | -     |
| SMI x TS x MAT               | 0.02           | 0.88   | -     |
| Temperature x SMI x TS x MAT | 5.01           | 0.03   | -     |

Table A2. Output ( $\chi^2$ - and p-values) of linear mixed effects models testing the effects of precipitation, mean annual temperature (MAT), soil moisture (SMI), and tree species (TS), on RWI. For main-effect terms, the variance inflation factor (VIF) is given (computed from the r-package *car* on a model containing only the main effects). Model sample size is 4578 RWI nested in 1979 plots. Phi-value of the AR1 temporal autocorrelation = 0.40. Plot intercept variance = 5.57x10<sup>-12</sup>. Tree nested within plot variance = 5.10x10<sup>-11</sup>.

| Term                | $\chi^2$ | p      | VIF   |
|---------------------|----------|--------|-------|
| Precipitation       | 217      | < 0.01 | 1.000 |
| SMI                 | 0.10     | 0.76   | 1.008 |
| TS                  | 0.01     | 0.92   | 1.006 |
| MAT                 | 1.50     | 0.22   | 1.013 |
| Precipitation x SMI | 24.3     | < 0.01 | -     |
| Precipitation x TS  | 371      | < 0.01 | -     |
| SMI x TS            | < 0.01   | 0.96   | -     |
| Precipitation x MAT | 1324     | < 0.01 | -     |
| SMI x MAT           | 0.28     | 0.60   | -     |
| TS x MAT            | 0.16     | 0.69   | -     |

| Precipitation x SMI x TS       | 4.84 | 0.03   | - |
|--------------------------------|------|--------|---|
| Precipitation x SMI x MAT      | 19.0 | < 0.01 | - |
| Precipitation x TS x MAT       | 10.9 | < 0.01 | - |
| SMI x TS x MAT                 | 0.02 | 0.90   | - |
| Precipitation x SMI x TS x MAT | 2.52 | 0.11   | - |

Table A3. Output (χ²- and p-values) of linear mixed effects models testing the effects of SPEI, mean annual temperature (MAT), soil moisture (SMI), and tree species (TS), on RWI. For main-effect terms, the variance inflation factor (VIF) is given (computed from the r-package *car* on a model containing only the main effects). Model sample size is 4578 RWI nested in 1979 plots. Phi-value of the AR1 temporal autocorrelation = 0.40. Plot intercept variance = 2.27x10<sup>-11</sup>. Tree nested within plot variance = 7.59x10<sup>-11</sup>.

| Term                  | $\chi^2$ | p      | VIF   |
|-----------------------|----------|--------|-------|
| SPEI                  | 1.12     | 0.29   | 1.000 |
| SMI                   | 0.10     | 0.76   | 1.008 |
| TS                    | 0.02     | 0.89   | 1.006 |
| MAT                   | 1.40     | 0.24   | 1.013 |
| SPEI x SMI            | 14.6     | < 0.01 | -     |
| SPEI x TS             | 42.5     | < 0.01 | -     |
| SMI x TS              | < 0.01   | 0.96   | -     |
| SPEI x MAT            | 712      | < 0.01 | -     |
| SMI x MAT             | 0.29     | 0.59   | -     |
| TS x MAT              | 0.14     | 0.71   | -     |
| SPEI x SMI x TS       | 0.22     | 0.64   | -     |
| SPEI x SMI x MAT      | 10.6     | < 0.01 | -     |
| SPEI x TS x MAT       | 67.7     | < 0.01 | -     |
| SMI x TS x MAT        | 0.02     | 0.89   | -     |
| SPEI x SMI x TS x MAT | 0.10     | 0.75   | -     |

Table A4. Output ( $\chi^2$ - and p-values) of linear mixed effects models testing the effects of VPD, mean annual temperature (MAT), soil moisture (SMI), and tree species (TS), on RWI. For main-effect terms, the variance inflation factor (VIF) is given (computed from the r-package *car* on a model containing only the main effects). Model sample size is 4578 RWI nested in 1979 plots. Phi-value of the AR1 temporal autocorrelation = 0.40. Plot intercept variance = 1.72x10<sup>-11</sup>. Tree nested within plot variance = 5.63x10<sup>-11</sup>.

| Term | $\chi^2$ | p      | VIF   |
|------|----------|--------|-------|
| VPD  | 326      | 

Figure A1. Correlation values among temporally dynamic explanatory climatic variables.

# Appendix C

Figure A2. Model output of the effects of precipitation on RWI (ring width index) when separated between low (10% quantile; blue lines), median (black lines), and high (90% quantile; red lines) MAT (mean annual temperature). Lines show the differences between low (10% quantile; solid line), and high (90% quantile; dashed line) SMI (soil moisture index). Facets show differences between species (left = P. abies; right = P. sylvestris). Note that quantiles represent different values for the different species. SMI values for P. abies (and P. sylvestris in parentheses) are 2.7 (1.9) and 85 (87), for low and high values, respectively. MAT values are 0.9 (1.1), 3.4 (3.7), and 7.4 (7.4), for low, median, and high values, respectively.

## Code availability

All code used in this study can be found at: https://github.com/LundgrenAndreas/Research/tree/main/Project GeoTree

## Data availability

Data on tree ring widths and forest characteristics are available for download at the Swedish National Forest Inventory website:

https://www.slu.se/en/Collaborative-Centres-and-Projects/the-swedish-national-forest-inventory/foreststatistics/microdatafor-download/ (see Fridman et al., 2014 for details on the dataset).

Data on climatic variables are available for download at the SMHI (Swedish Meteorological and Hydrological Institute) website: https://www.smhi.se/data/utforskaren-oppna-data/meteorologisk-ateranalys-smhigridclim-uerra-harmonie (see Andersson et al., 2021 for details on the dataset).

Data on modeled soil moisture are available for download at: https://www.slu.se/en/departments/forest-ecology-management/forskning/soil-moisture-maps/here-are-the-maps/.

All data used in this paper (except site coordinates) are available at: https://doi.org/10.5281/zenodo.12655494 (Lundgren et al., 2025).

## **Author contributions**

- AL: Conceptualization, methodology, investigation, validation, writing original draft, project administration, and funding acquisition.
  - JS: Conceptualization, methodology, validation, writing review and editing, project administration, and funding acquisition.
  - JE: Conceptualization, methodology, validation, writing review and editing.
  - GG: Conceptualization, methodology, validation, writing review and editing, and funding acquisition.

## 475 Competing interests

The authors declare that they have no conflict of interest.

## Acknowledgments

We thank the Swedish National Forest Inventory (Riksskogstaxeringen) for data acquisition.

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
