# Peer review of "Unpacking climate effects on boreal tree growth: An analysis of treering widths across temperature and soil moisture gradients"

_EGUsphere, 2025_

## Author Comment (AC1)

**We thank the reviewer for the valuable comments and suggestions, and for taking the time to help us improve our manuscript! Please see our answers in bold below each comment.**

General comments

How are the results relevant for simulation models of tree growth and competition (that is, for projections of climate change effects on forests at landscape and broader scales) especially for those models that account for physiological species traits such as temperature and drought tolerance?

**Unfortunately, we do not work with large-scale, process-based forest/ecosystem models. As a result, it is difficult for us to know how our findings are relevant to such modelling work. Maybe our results can be used for qualitative comparisons in simulations using process-based forest models (e.g. CLIM4, LPJ-GUESS, LANDIS-II). For example, significant discrepancies in growth trends between our results and models across climate gradients may reveal gaps or misspecifications in process-based models. Nonetheless, we believe that additional feedback from the reviewer is needed for us to answer this question.**

I assumed that the methodology is sound (not my expertise) and concluded that the study is sound and the results useful. I found the study interesting, well-presented, and I think it would make a great addition to the forestry literature.

In general, the English is understandable and quite passable, but it could be improved. I have made some suggestions below, but others could be made. However, the English is very adequate given that English is not likely the native language of any of the authors.

**During our revision, we will conduct a thorough proofreading to improve the language.**

Specific comments

First paragraph. The phrase "at stake" as used here is probably not the best because it is somewhat ambiguous. "At risk" or "threatened" would be more accurate.

**We will change the phrasing to read "[…] at risk."**

L 52. Warming may also affect disturbance rates and intensity, also impacting mortality. See https://doi.org/10.1016/j.scitotenv.2024.177043.

**Yes, in our analyses we do not account for mortality, and we mention this in the discussion. We will elaborate further and add the suggested reference as an example that mortality and ecological interactions need to be addressed to fully understand the outcomes of climate change.**

L 93.  What is the expected mechanism driving the prediction about species responses in H3?  Such mechanisms are important to convey the a priori nature of your hypotheses.

**We will expand the section starting at L72 to explain a potential mechanism in the different responses between the species. Namely that a higher root:leaf ratio, previously found in Pinus sylvestris, may provide greater access to water.**

L 102.  Is this describing the design of the inventory or your study?  Or both?  Not clear.

**We will clarify that this is the design of the NFI inventory program.**

L 109.  Give the rationale for excluding these trees.  Also provide the rationale for the method described in L 100.

**For L109, we will clarify that we excluded trees in wetlands to focus our study on trees growing in productive forest landscapes. Regarding L100, it is not clear to us what 'method' we should provide more information on. Should we expand on the design for data collection throughout Sweden, or should we elaborate on the rationale for using radial growth as a response?**

L 112.  Reword to: "value magnitudes higher than"

**We will change the text in accordance with the comment.**

L 115.  Define acronym (COFECHA).

**We will clarify that COFECHA is a dendrochronological software. However, COFECHA is not an acronym, it's an invented Spanish word for co-date or cross-date, so we will therefore not add any further definition.**

L 123.  What time period do the climate data represent?

**We will add the period that the climate data represent (1961-2018).**

L 138.  Reword to: "could not end until after September 1st."

**We will change the text in accordance with the comment with the slight alteration of "could not end until after August 31st".**

L 143.  Run-on words.

**We will restructure the sentence to read "The modeled data consist of values ranging from 0 to 100 at a 2 m resolution raster grid, where values indicate the probability of being classified as the "wet" category in the NFI inventory field plots." Hopefully this will be more easily read.**

L 144.  Is "mean soil moisture values" the wetness probability described in the prior sentence? Unclear.

**We will change the sentence to clarify that it indeed indicates the data introduced in the prior sentence. The sentence will instead read "Using the modeled soil moisture data, we calculated a soil moisture index (SMI) based on mean soil moisture values in a buffer of 25 m radius around individual trees using zonal statistics in QGIS (QGIS Association, 2021)."**

L 184.  Re: "while precipitation and SPEI…" – should this read "reduced precipitation"?

**We will clarify that it is increased precipitation that has a generally negative effect.**

L 188.  Re: "The growth response to precipitation and SPEI became increasingly positive with increasing MAT (Fig. A2; Table 1). Has the SPEI acronym been defined?  I think the index itself should be defined to ensure that readers are aware that it goes quite negative with severe drought.

**At the introduction of SPEI (in the section "Climate and soil moisture data"), we will clarify the range of SPEI. The new sentence will read "SPEI centers around 0, where negative values indicate drier conditions and positive values indicate wetter conditions."**

Results presentations (especially graphs) are excellent!  The results data are difficult to interpret, but the presentation helps some and the text descriptions of the meaning of the results help a LOT.

L 260.  Here and throughout, you might consider using "relative MAT" given that in Sweden, your MAT is relatively low compared to elsewhere on the globe.  You might somewhere in the discussion discuss the generality of your results in the context of the globe.

**This is a good point. However, we refrain from using "relative MAT" as this can be interpreted as the temperature being a relative measure. Regarding the generality of our results in the context of the globe, we will elaborate on the geographical range of similar findings and add a sentence referring to similar patterns (in terms of increasingly negative growth-responses to temperature along MAT gradients) in tropical and temperate forests. The new sentence will read "Although boreal forests experience relatively low temperature, similar patterns have also been observed in the tropical (Zuidema et al., 2022) and temperate (Charru et al., 2017) biomes."**

L 277.  I would have liked some speculation about the mechanism for this result.  Is permafrost involved?  Is some precipitation in the form of snow that is lost before the growing season?

**We will add a section on potential mechanisms regarding the negative growth-response to precipitation. The new suggested text reads "A potential mechanism of the negative effects of precipitation may be that higher amounts of snowfall delays the start of the growing season (D'Orangeville et al., 2016). However, the negative response in our study is based on growing season precipitation sums only, thus likely excluding snowfall. Another possible reason as to why**

**precipitation negatively affects tree growth in the colder regions is that the forests are already near water saturation and excess precipitation causes waterlogging (Laudon et al 2024).”**

L 299.  Didn't you exclude plots on wet sites?

**It is true that we excluded plots in wetlands. However, the trees of our study grow in a range of relatively wet to relatively dry conditions. Hence, the pattern seen in previous studies (where trees growing in wet areas (not necessarily wetlands) are less affected by drought conditions than those growing in dry areas) could reasonably be thought to exist in our gradient as well. However, we will add some text to remind readers that wetland trees were excluded. The new section will read “Notably, we excluded trees growing in wetlands in our study. The inclusions of such trees may have revealed a greater effect of soil moisture on the trees' growth response to drought conditions.”**

L 321.  I would like to see more exploration of this discrepancy with other results.  Did this cause you to question your results?  On what basis do you trust these results?

**We found the minor effect of soil moisture somewhat surprising, and we have done further analyses to validate the results. We have run the analyses on soil moisture indices based on different buffer zones (from 1 to 1000 meter buffer zones around each tree) and found similar results regardless of the buffers used. Furthermore, we have made use of the NFI field estimates of the soil moisture (classified as wet, moist, mesic, dry) and explored RWI correlations to temperature and precipitation, and found no effects of soil moisture. Hence, both modeled and field estimated data suggest that soil moisture has a very limited effect on the tree growth-response to climatic variables.**

L 328.  "indicators for" seems to be the wrong phrase here.  "drivers of?"

**We will change the text in accordance with the comment.**

L 337.  Other studies have suggested that long periods of stress are required to actually kill trees because even one good year can rebuild reserves.  For example, see DOI: 10.1002/ecs2.1253.

**This is a good point and we will expand our caveats for the extreme year analysis by including the following sentence: “It is also important to note that we have studied discrete extreme years, but it may be extended periods of extreme conditions rather than single-year extremes that produce the most severe effects for tree growth (Gustafson et al., 2016).”**

---

## Author Comment (AC2)

Tree growth is a fundamental and widely used term in plant science and ecology. The current study aimed to explore and explain tree ring width response to temperature and precipitation gradient over almost 2000 plots across a gradient of climatic conditions in the boreal forests of Sweden. The analysis revealed that in cold environments tree growth responds positively to an increase in the ambient temperature, while in "warm" environments, trees reduce growth due to an increase in ambient temperature. Additionally, soil moisture can mitigate reduction in tree growth in areas of high MAT. Overall, the analysis is well done and has an important value. However, as a whole, this study needs to strength and clarify the main message. I found some descriptions are too detailed, whereas some important steps were either ignored or provided with very little information. Therefore, I recommend this to be gone through a major revision. I explain my concerns in more detail below.

Major comments:

1.  The ideas in introduction are leaping through one to another without a proper flow. Also, different terms were not consistent across the document, which makes this document very confusing. It begins with a description of the economic implications of tree growth, which is not the central focus of the study. Additionally, there is excessive use of the term "climate change" and occasional use of "global warming." I recommend define the scope of what aspect of climate change is being investigated. This needs to be clearly defined, as "climate change" on its own is too vague.

    Terms like "high temperature" and "high/low soil moisture" are also imprecise and need to be quantified or clearly contextualized—what temperature or soil moisture levels are considered "high"? use the actual value or range of values. That will allow a more application result in other areas. Overall, this section failed to answer/ satisfy "why" it is important? Then, need more specific information about "why" this particular species?

**Regarding the flow of the text: We will make an effort to improve flow and language. Regarding the inconsistencies in the terms: We will work through the text and try to sort out any inconsistencies.**
**Regarding the description of economic implications: This is one of several ecosystem services that we mention in the introduction. As the reviewer later implies, it is important to explain why the study is important, and we believe that the early mention of ecosystem services that may be at risk by reduced tree growth should aid in producing such an explanation.**
**Regarding the terms "climate change and global warming": We will change the text to clarify that it is changes in temperature and precipitation that are of interest to our study.**
**Regarding the "high"/"low" terms: We will add a section in the methods where we introduce our study area. Here, we will also define the temperature/precipitation ranges and how these have changed over time. By adding ranges of tested variables, it should be clear to the reader where on the scale high/low refers to. This can also be viewed in Figure 1. Soil moisture is not in a form that directly can be translated to standardized metrics such as volumetric soil moisture. We will clarify this in the methods when describing the soil moisture variable used in our study.**
**Regarding the "why is this important": We are not sure what the reviewer refers to here. "Why" this study or "why" these species? In any case, we think it will be clearer in the revised version of the introduction where we in more details will explain the ecosystem services provided by the boreal forests and the importance that the studied tree species have for the delivery of these services.**

2. The tree species (the core subject of the paper) are barely introduced. The introduction does not explain why these two species are meaningful. The introduction should describe their geographical distribution, ecological characteristics, and the typical range of temperature, VPD, and soil moisture conditions in which they thrive. It should also reference recent studies related to their responses to environmental stressors that are investigated here.

**We do not agree that the tree species is the core subject of the study and therefore have not expanded on this more than the other subjects (i.e. local temperature and soil moisture conditions). However, we will add information (in the introduction) about why we hypothesize species differences in growth-climate responses. Furthermore, in the introduction as well as in a new paragraph on site description within the methods section, we will clarify that the studied tree species are the dominant species within our study area, which justifies the choice to focus our analyses on these specific tree species.**

3. All the hypotheses refer to specific regions that are not even mentioned in the introduction, despite playing a central role in the study.

**Regarding the hypotheses: This was an unfortunate use of the word region. What we mean are areas within our geographical range that experience specific conditions, for example, higher/lower temperature compared to other areas. We will rephrase these hypotheses to avoid such misunderstanding.**

**To ensure that readers understand the scope of our study area, we will also begin the methods section by explaining the geographical range of our study, as well as how some variables vary across our sites (provided ranges of e.g. mean temperature and length of snow period different parts of our study area). We believe that this will put the study region in a better context compared to other geographical areas.**

4. There is a clear lack of background on previous research on the physiological responses of conifer trees to temperature and soil moisture. Key studies in the field—such as those by Wagner et al., 2021; Zweifel et al., 2006, Zweifel et al., 2021; Klein et al., 2014; Novick et al., 2024 —are notably absent and should be incorporated to establish the scientific context on trees physiological response to soil and atmospheric conditions.

**We will add Wagner et al 2022 to the description on the physiological processes that cause droughts to affect trees. We will add Zweifel et al 2006 to the discussion on the lack of effects from soil moisture. We will add Klein et al 2014 to the introduction to extreme events linking drought conditions to mortality events. We will add Novick et al 2024 to the introduction regarding the negative effects of VPD and to the discussion on the interaction between rising temperature and soil moisture conditions.**

5. H4 needs to be rewritten and detailed- how each species will respond to an increase/ decrease in ambient temperature? How will each species respond to an increase/decrease in soil moisture? And add a paragraph in the introduction that explains why you hypothesize for each species.

**We will specify our hypothesis for each species and clarify how we expect them to respond to change along the gradients within our study area. In the introduction, we will also expand on**

**the reasoning behind why we expected differences between the species (they have different ecological niches).**

6. In the method section, a brief sites descriptions was missing, more details, such as what the dominant tree species are, their approximate age, and overall environmental conditions. Snow? Solar radiation?

**We will add a site description at the beginning of the methods section, including the dominant tree species, and environmental conditions (temperature, precipitation, snow cover).**

7. In Figure 1B, the comparison between high and low soil moisture needs to be clearly defined. What thresholds or criteria were used to distinguish "high" from "low" soil moisture? And why not give the actual values of soil moisture (v/v %)?

**The reviewer makes a good point and we will add a sentence in the figure text explaining the colour gradient (i.e. that yellow or "Low" indicates a 0-20% probability of the site being classified as "wet" based on the modelled data we use in this study). Unfortunately, we cannot provide actual values of soil moisture since the data we use is modelled based on an index that use topography and ground water data and produces values that correspond to a probability of being classified as "wet" in a categorical gradient. Details on this soil moisture index were unfortunately not included in the method section in the earlier version, but will be included in the revised version.**

8. Regarding the calculation of the Standardized Precipitation-Evaporation Index (SPEI), the authors mention using net solar radiation based on latitude. However, cloud cover, which can significantly affect incoming radiation, is not addressed. Was this factor considered or accounted for in the calculations?

**This is a good point. Although SPEI is often calculated with latitude as a proxy for radiation, regional differences in cloud cover can skew this calculation. However, in our region of study (Sweden), the irradiance measurements over the last 30 years from satellite data and meteorological stations suggest a relatively coherent drop in irradiance ($kWh/m^2$) with latitude, and we deemed that this variation would be of very limited importance for our results.**

9. In the results section, I suggest adding a short description of the number of extreme events experienced by the trees during the study period. Were high-temperature extremes or low-precipitation events more frequent? Additionally, it would strengthen the results to quantify the reduction in RWI associated with extreme increases in temperature and extreme reductions in precipitation.

**In this study, we have defined "extreme years" as the 10% highest temperature years (or 10% lowest precipitation years) similarly to the definition in Rammig et al 2015. Since our data on both temperature and precipitation have been modelled through 60 years, there are 6 extreme years in both cases for each site.**
**Regarding the quantification of RWI reduction, we have used "Resistance" values to show the reduction/increase in RWI during extreme years.**

10. I recommend expanding the discussion on the threshold between atmospheric demand (VPD) and soil moisture availability. This would provide valuable ecological insight into species-specific sensitivities under climatic stress.

**We will expand the section regarding the interaction between soil moisture and temperature (and VPD), mainly with information provided from the Novick et al 2024 review that the reviewer has suggested.**

---

## Author Response (AR1)

Listed below are our changes to the manuscript after considering the reviewers' comments. Note that the indicated lines refer to the complete manuscript (not the track changes version). Apart from the changes suggested by the reviewers, we have also made a change to our statistical approach where we now have used Wald Chi-square tests to test the effects of our models rather than simply summarising the output of the models. We have added a line explaining this in the methods section (L 218) and updated all tables of statistics. Despite slight changes to the statistical output, the general results and our conclusions based on those results remain the same.

**Comments from reviewer 1**

How are the results relevant for simulation models of tree growth and competition (that is, for projections of climate change effects on forests at landscape and broader scales) especially for those models that account for physiological species traits such as temperature and drought tolerance?

While we have made no changes in relation to the initial comment on models of tree growth and competition (see response to reviewer comment), we have made an effort to discuss how our results may be generalized to broader perspectives in accordance with the reviewer's follow-up comment. For example, see discussion at line 301-306 for a comparison with similar studies in other biomes (tropical and temperate). Further, at line 317-322, we discuss our result in relation to changed precipitation patterns (both in terms of snow during the non-growing season and rain during the growing season). In section 4.2, starting at L 342, where we discuss the tree growth response in relation to local soil moisture, we have expanded the discussion on the potential mechanisms that might explain the surprisingly week effect of local soil moisture. By this we hope to provide a better context for the important message that climate change adaptation of forest management needs to acknowledge that the importance of soil moisture will differ depending on whether you consider the southern or northern boreal region. Finally we have expanded the conclusion paragraph to broaden our results to perspectives important for forest management (L 407-414).

In general, the English is understandable and quite passable, but it could be improved. I have made some suggestions below, but others could be made. However, the English is very adequate given that English is not likely the native language of any of the authors.

We have gone through the text to improve the language (see the track changes version).

First paragraph. The phrase "at stake" as used here is probably not the best because it is somewhat ambiguous. "At risk" or "threatened" would be more accurate.

We have changed the text in accordance with the comment. See L 33.

L 52. Warming may also affect disturbance rates and intensity, also impacting mortality. See https://doi.org/10.1016/j.scitotenv.2024.177043.

We have expanded our introduction of extreme disturbances and added the suggested reference to section of increased frequency and intensity of disturbances and the link to tree mortality. See L 53.

L 93. What is the expected mechanism driving the prediction about species responses in H3? Such mechanisms are important to convey the a priori nature of your hypotheses.

We have expanded on the species-section of the introduction where we provide a potential mechanism for difference between the studied species: that a higher root:leaf ratio, previously found in Pinus sylvestris, may provide greater access to water and thereby greater drought resistance. See section starting at L 78.

L 102. Is this describing the design of the inventory or your study? Or both? Not clear.

We have clarified that this is the design of the NFI inventory program. See L 132.

L 109. Give the rationale for excluding these trees. Also provide the rationale for the method described in L 100.

Regarding the first rationale, we have clarified that we excluded trees in wetlands to focus our study on trees growing in productive forest landscapes. See L 139.

Regarding the second rationale, we have not made any changes (see discussion with reviewer 1).

L 112. Reword to: "value magnitudes higher than"

We have changed the text in accordance with the comment. See L 143.

L 115. Define acronym (COFECHA).

We have clarified that COFECHA is a dendrochronological software. See L 147.

L 123. What time period do the climate data represent?

We have clarified that the climate data represent the period 1961-2018. See L 158.

L 138. Reword to: "could not end until after September 1st."

We have reworded to "could not end until after August 31st". See L 172.

L 143. Run-on words.

We have reworded the sentence to read: "The modeled data consist of values ranging from 0 to 100 at a 2 m resolution raster grid, where values indicate the probability of being classified as the "wet" category in the NFI inventory field plots." See L 177.

L 144. Is "mean soil moisture values" the wetness probability described in the prior sentence? Unclear.

We have reworded the sentence to clarify that it is indeed the wetness probability previously described. See L 179.

L 184. Re: "while precipitation and SPEI..." – should this read "reduced precipitation"?

We have clarified that it is increased precipitation that has a generally negative effect. See L 223.

L 188. Re: "The growth response to precipitation and SPEI became increasingly positive with increasing MAT (Fig. A2; Table 1). Has the SPEI acronym been defined? I think the index itself should be defined to ensure that readers are aware that it goes quite negative with severe drought.

At the introduction of SPEI (in the section "Climate and soil moisture data"), we have now clarified the range of SPEI. See L 162.

L 260. Here and throughout, you might consider using "relative MAT" given that in Sweden, your MAT is relatively low compared to elsewhere on the globe. You might somewhere in the discussion discuss the generality of your results in the context of the globe.

We have added a section on site description in the methods where general temperature and precipitation patterns in the study area (Sweden) are introduced. Hopefully this will clarify how our study area compares to global patterns. See L 112.

In the discussion, we have added references from tropical and temperature forest ecosystems to compare our findings with these biomes. See L 301.

L 277. I would have liked some speculation about the mechanism for this result. Is permafrost involved? Is some precipitation in the form of snow that is lost before the growing season?

We have added a section with two potential mechanisms regarding the negative growth-response to precipitation: Growing season delay due to snowfall; and Excess precipitation causing waterlogging. See L 317.

L 299. Didn't you exclude plots on wet sites?

We have added a reminder about the exclusion of wet sites in the discussion, as well as a caveat that their inclusion could change our results. See L 346.

L 321. I would like to see more exploration of this discrepancy with other results. Did this cause you to question your results? On what basis do you trust these results?

We have made no changes based on this comment (see response to reviewer comment).

L 328. "indicators for" seems to be the wrong phrase here. "drivers of?"

We have changed the text in accordance with the comment. See L 381.

L 337. Other studies have suggested that long periods of stress are required to actually kill trees because even one good year can rebuild reserves. For example, see DOI: 10.1002/ecs2.1253.

We have added a caution, coupled with the suggested reference, that our extremes are based on single-year values and that extended periods of extreme conditions may have more severe effects. See L 395.

Comments from reviewer 2

Tree growth is a fundamental and widely used term in plant science and ecology. The current study aimed to explore and explain tree ring width response to temperature and precipitation gradient over almost 2000 plots across a gradient of climatic conditions in the boreal forests of Sweden. The analysis revealed that in cold environments tree growth responds positively to an increase in the ambient temperature, while in "warm" environments, trees reduce growth due to an increase in ambient temperature. Additionally, soil moisture can mitigate reduction in tree growth in areas of high MAT. Overall, the analysis is well done and has an important value. However, as a whole, this study needs to strength and clarify the main message. I found some descriptions are too detailed, whereas some important steps were either ignored or provided with very little information. Therefore, I recommend this to be gone through a major revision. I explain my concerns in more detail below.

**Major comments:**

1. The ideas in introduction are leaping through one to another without a proper flow. Also, different terms were not consistent across the document, which makes this document very confusing. It begins with a description of the economic implications of tree growth, which is not the central focus of the study. Additionally, there is excessive use of the term "climate change" and occasional use of "global warming." I recommend define the scope of what aspect of climate change is being investigated. This needs to be clearly defined, as "climate change" on its own is too vague.

Terms like "high temperature" and "high/low soil moisture" are also imprecise and need to be quantified or clearly contextualized—what temperature or soil moisture levels are considered "high"? use the actual value or range of values. That will allow a more application result in other areas. Overall, this section failed to answer/ satisfy "why" it is important? Then, need more specific information about "why" this particular species?

We have made an effort to improve the flow and language of the text (see the track changes version).

We have clarified that it is changes in temperature and precipitation that we study, rather than using the general term "climate change".

We have added a section on site description in the methods to clarify what "high" and "low" temperature/precipitation refers to in our study. See L 112.

We have added a more detailed description in the section on soil moisture data to clarify what the soil moisture variable indicates in our study. See L 176.

The tree species (the core subject of the paper) are barely introduced. The introduction does not explain why these two species are meaningful. The introduction should describe their geographical distribution, ecological characteristics, and the typical range of temperature, VPD, and soil moisture conditions in which they thrive. It should also reference recent studies related to their responses to environmental stressors that are investigated here.

In the introduction, we have expanded on the description of the tree species used in our study, as well as provided a basis for our hypothesis that the two species will differ in their growth-climate responses. See L 78.

In the new site description section in the methods, we have added a paragraph on the tree species' distributions through our study area. See L 113.

2. All the hypotheses refer to specific regions that are not even mentioned in the introduction, despite playing a central role in the study.

We have reworded the hypotheses to clarify that we indicate regions within our study area (which is presented in the prior sentence). For example, we have added an explanatory sentence to ensure that regions with high MAT correspond to the southern part of our study area. See L 99.

The new site description should further put the study region in a better context compared to other geographical areas. See L 112.

3. There is a clear lack of background on previous research on the physiological responses of conifer trees to temperature and soil moisture. Key studies in the field—such as those by Wagner et al., 2021; Zweifel et al., 2006, Zweifel et al., 2021; Klein et al., 2014; Novick et al., 2024 —are notably absent and should be incorporated to establish the scientific context on trees physiological response to soil and atmospheric conditions.

We have added Zweifel et al 2006 as a reference to the discussion on the minor effects from soil moisture. See L 349.

We have added Klein et al 2014 to the introduction to extreme events linking drought conditions to mortality events. See L 58.

We have added Novick et al 2024 to the introduction regarding the negative effects of VPD and to the discussion on the interaction between rising temperature and soil moisture conditions. See L 43 and L 354.

4. H4 needs to be rewritten and detailed- how each species will respond to an increase/ decrease in ambient temperature? How will each species respond to an increase/decrease in soil moisture? And add a paragraph in the introduction that explains why you hypothesize for each species.

We have specified our hypothesis for each species and clarified how we expect them to respond to change along the gradients within our study area. Namely that *P. sylvestris* will have a more positive response to increased temperature in cold areas, and more negative response in warm areas, than *P. abies*. See L 104.

We have also added references to justify our reasoning in the introduction of the species. See L 78.

5. In the method section, a brief sites descriptions was missing, more details, such as what the dominant tree species are, their approximate age, and overall environmental conditions. Snow? Solar radiation?

We have added a site description at the beginning of the methods section, including the dominant tree species, and environmental conditions (temperature, precipitation, snow cover). See L 112.

6. In Figure 1B, the comparison between high and low soil moisture needs to be clearly defined. What thresholds or criteria were used to distinguish "high" from "low" soil moisture? And why not give the actual values of soil moisture (v/v %)?

We have expanded on the figure caption to provide a more detailed explanation of the color gradient, e.g. that yellow or "Low" indicates a 0-20% probability of the site being classified as "wet" based on the modelled data we use in this study. See L 151.

We have expanded the explanation of the soil moisture index that we have used, to clarify that we do not use data on volumetric soil moisture content. Hopefully this will avoid confusion about what the soil moisture values indicate. See L 177.

7. Regarding the calculation of the Standardized Precipitation-Evaporation Index (SPEI), the authors mention using net solar radiation based on latitude. However, cloud cover, which can significantly affect incoming radiation, is not addressed. Was this factor considered or accounted for in the calculations?

We have made no changes based on this comment (see response to reviewer comment).

8. In the results section, I suggest adding a short description of the number of extreme events experienced by the trees during the study period. Were high-temperature extremes or low-precipitation events more frequent? Additionally, it would strengthen the results to quantify the reduction in RWI associated with extreme increases in temperature and extreme reductions in precipitation.

We have made no changes based on this comment (see response to reviewer comment).

9. I recommend expanding the discussion on the threshold between atmospheric demand (VPD) and soil moisture availability. This would provide valuable ecological insight into species-specific sensitivities under climatic stress.

We have expanded the section regarding the interaction between soil moisture and temperature (and VPD), mainly with information provided from the Novick et al 2024 review that the reviewer has suggested. See L 352.

---

## Author Response (AR2)

1) in the new section in Methods (2.1 site description) you mention that there have been changes in the precipitation regime of the study area. It is not clear whether these changes are statistically significant or not, and whether you performed those analyses yourself or you just extracted the data from SMHI 2025. Please clarify this in the text.

We have clarified that these patterns have been observed through meteorological observations carried out by SMHI. We have also added the reference to SMHI 2025 on each climatological section in the site description (Temperature, Precipitation, Snow cover) to further clarify that these patterns are based on SMHI's observations.

2) I see that the information provided by the new tests performed (e.g., Table 1) include p-values that are <= 0.01 (smaller or equal to). Please, clarify whether they are smaller or equal to 0.01 (actually, another decimal would be preferable). Alternatively, is there a good reason why they have been reported as smaller or equal to 0.01?

We have added a third decimal to all p-values. Note that almost all values that were previously reported as "< 0.01" are now reported as "< 0.001" since the p-values for many effects are very low (e.g.  $< 1 \times 10^{-16}$ ).